# A Formal Model of Concurrent Planning and Execution with Action Costs

**Ava Bensoussan,[1] Eyal Shlomo Shimony,[1] Shahaf S. Shperberg,[1,2] Wheeler Ruml[3]**

[1]Ben-Gurion University of the Negev, Israel
[2]University of Texas at Austin, USA
[3]University of New Hampshire, USA
avabe@post.bgu.ac.il, shimony@cs.bgu.ac.il, shperbsh@post.bgu.ac.il, ruml@cs.unh.edu

## Abstract

If a planning agent is considering taking a bus, for example, the time that passes during its planning can affect the feasibility of its plans, as the bus may depart before the agent has found a complete plan. Previous work on this situated temporal planning setting proposed an abstract deliberation scheduling scheme for minimizing the expected cost of a plan that is still feasible at the time it is found. In this paper, we introduce the CoPEC model, extending the deliberation scheduling approach to allow for executing initial actions before the solution plan is fully specified. Although this may allow lower overall time to goal achievement, it also entails the risk of performing incorrect actions, a tradeoff formally treated in our model. We develop algorithms to solve CoPEC instances. Our empirical evaluation shows that a greedy scheme performs well in practice on problem instances generated from weighted 15-puzzle instances.

## 1 Introduction

Situated temporal planning (Cashmore et al. 2018a) is a model for the planning problem faced by an agent for which significant time passes as it plans. In this setting, external temporal constraints (e.g., deadlines) are introduced, depending on the actions in a plan. For example, taking the 9:00 bus introduces a new constraint that the agent must be at the bus stop by 9:00. These plan-specific constraints make the problem different from real-time search (e.g., (Koenig and Sun 2009; Sharon, Felner, and Sturtevant 2014; Cserna, Ruml, and Frank 2017; Cserna et al. 2018)), deadline-aware search (Dionne, Thayer, and Ruml 2011), or Utility Guided Search (Burns, Ruml, and Do 2013).

Situated temporal planning calls for a search strategy different from traditional offline search algorithms, as the choice of which node to expand must account for the fact that time spent exploring one part of the search space passes in the real world and may invalidate some partial plans. Shperberg et al. (2019) suggested a rational metareasoning (Russell and Wefald 1991) scheme for situated temporal planning. They formalized the problem as an MDP called $(AE)^2$ (for Allocating Effort when Actions Expire) whose actions allocate a unit of time to one of $n$ running processes, showed that solving this MDP optimally is NP-hard, and suggested a greedy decision rule (denoted as P-Greedy) that worked well empirically. However, P-Greedy

attempts merely to maximize the chance of finding a timely plan, without considering plan cost. For example, if taking a taxi does not introduce a deadline but is much more expensive than taking the bus, P-Greedy always chooses to take the taxi, even if there is very little uncertainty about whether the agent could catch the bus on time. Factoring in solution cost was done in Shperberg et al. (2020b), by introducing the $(ACE)^2$ model (Allocating Computational Effort when Actions (with Costs) Expire), defining the appropriate MDP, and introducing a greedy algorithm for assigning processing time in order to minimize expected cost.

All these prior works assume that planning must terminate before starting to act, and since planning takes time, this may cause otherwise valid plans to become invalid. One way to ameliorate this problem is by starting to act before the plan is complete, although obviously this entails the risk of performing actions that will cause failure. For example, consider an agent that needs to get to the airport, and can do so by riding the train or by taking a taxi. Both of these options can be thought of as partial plans that need to be elaborated into complete plans. Now, suppose that the estimated time to completely refine each plan is seven minutes, but that both plans expire in six minutes. Suppose also that the planner has already determined that the first action in the train plan is to walk to the station, which takes three minutes, and that the first action in the taxi plan is to call a taxi, which takes two minutes. The only way to achieve the agent's goal in time is to start acting before a complete plan is formulated.

In this paper, we extend the $(ACE)^2$ model to allow for concurrent planning and execution. First, we provide a formal model of the resulting tradeoff between risk and rewards in executing actions before planning completes, called CoPEC (Concurrent Planning and Execution with action Costs), and define an appropriate MDP of this problem. We examine theoretical properties of this model, and point out indicators of intractability that go beyond that of the $(ACE)^2$ model. Due to the difficulty of exact solution, we examine special cases and provide an analytical solution to the special case where only one process remains. This solution is used to construct a greedy decision rule for the general case. Our empirical evaluation suggests that the new greedy schemes perform significantly better than various baseline algorithms and the P-Greedy scheme on benchmarks based on distributions gathered from the weighted 15-

puzzle. This ongoing work provides a formal foundation for the central issue in concurrent planning and execution.

## 2 Background

Cashmore et al. (2018b) formulated a situated temporal planning problem as propositional temporal planning with Times Initial Literals (TILs) (Cresswell and Coddington 2003; Edelkamp and Hoffmann 2004), which is specified by a tuple $\Pi = \langle F, A, I, T, G \rangle$. $F$ is a set of Boolean propositions that describe the state of the world. $A$ is a set of durative actions. Each action $a \in A$ has a duration in the range $[dur_{min}(a), dur_{max}(a)]$, and a start, invariant and end conditions (all are subsets of $F$). Each action $a$ also has a start and end effect, which specify which propositions in $F$ are affected when $a$ starts or ends. $I \subseteq F$ specifies the initial state, and $G \subseteq F$ specifies the goal condition. $T$ is a set of timed initial literals (TILs), each TIL $l$ affects a proposition $lit(l) \in F$ at time $time(l)$. A solution to a situated temporal planning problem is a sequence $\sigma$ of triples $\langle a, t, d \rangle$ (i.e. a schedule), where $a \in A$, $t$ is the time at which the execution of $a$ starts, and $d$ is the duration of the execution of $a$. This work abstracts away from details of the state or action representation and search process. For example, rather than considering TILs themselves, we assume possible deadlines, based on the TILs, for each action $a \in A$. These deadlines may or may not be known in advance, but as in $(AE)^2$ and $(ACE)^2$, we assume a known distribution over deadlines.

In $(AE)^2$ and its extended version $(ACE)^2$ (Shperberg et al. 2020a) that includes plan costs, one posits the existence of $n$ computational processes, all attempting to solve the same problem. One can imagine these representing partial plans that are competing to be elaborated. $(ACE)^2$ introduced a cost of failure $c_f$, and the problem was to minimize the cost of a timely plan executed after it was found by some process that has terminated; with a cost of $c_f$ if no such plan is found in time. For every process $i$ there is also a possibly unknown deadline, after which the solution cannot be used. The following distributions are assumed to be known in the $(ACE)^2$ model: **(i)** $D_i(t)$, the cumulative distribution function (CDF) over wall clock times of a random variable denoting the deadline for each process $i$; **(ii)** $M_i(t)$, the CDF giving the probability that process $i$ will terminate when given an accumulated computation time of $t$ or less; and **(iii)** $C_i$, a probability mass function (PMF) over solution costs for process $i$. The true values of the deadline and the plan cost for process $i$ are revealed only when process $i$ terminates. The cost of a plan of a process that has not terminated, as well as the cost of a terminated process which failed to deliver a solution, is $c_f$.

The objective of the $(ACE)^2$ model is to schedule processing time between the $n$ processes, optionally stopping deliberation and executing a complete plan delivered by one of the processes, so as to minimize the expected cost of the executed plan. It is necessary to include an explicit decision to start executing a plan, because even after a timely plan is found, we may want to keep searching for a plan with a lower cost.

Shperberg et al. (2020a) then formulated a discrete-time version of the problem, called $D(ACE)^2$, which allowed them to model the problem as an MDP. But since allowing dependencies in the model leads to severe intractability (PSPACE-complete), all distributions were assumed to be independent in the subsequent analysis.

**Greedy Scheme for** $(ACE)^2$  Shperberg et al. (2020a) used their analysis of the special case of $(ACE)^2$, where only one process has not terminated, to propose a greedy scheme for the general case. They called that greedy scheme Delay-Aware Greedy (DAG). Therein, they defined a policy $\pi_{i,k}(t, t_d)$ that allocates at most $t$ time to process $i$, starting after delay $t_d$, always stopping computation at time $d_k$, at which time the least-cost available plan is executed. The expected cost of such a policy was denoted by $E_{\pi_{i,k}}(t, t_d)$ and can be computed in linear time. For only one unterminated process, just setting $t_d$ to 0 and $t$ infinite, and maximizing over $k$, we get the optimal policy.

But when other processes can also be scheduled, there is a tradeoff, so they defined the most-effective reward gain (i.e. cost reduction) rate for process $i$, relative to the current best valid plan cost $c_c$ as:

$$ecr_i(t_d) = \max_{t,k} \frac{c_c - E_{\pi_{i,k}}(t, t_d)}{t} \qquad (1)$$

The value $ecr_i(0)$ represents the highest returns rate (minus expected cost). However, some processes are more time-critical than others. This can be measured by how much the returns rate decreases due to delaying the time at which the process starts running. Therefore, they defined the following criterion, which trades off high returns rate and decrease in returns rate due to delay.

$$Q_i(t_d) = ecr_i(0) - \gamma ecr_i(t_d) \qquad (2)$$

where $\gamma$ is an empirically determined constant that is used to balance immediate reward and future loss. The DAG scheme allocates time to the process $i$ that maximizes $Q_i(t_d)$ (Equation 2). Note that $t_d$ was empirically determined as well.

## 3 The CoPEC Model

### 3.1 Problem Statement

This paper extends the $(ACE)^2$ model to allow concurrent planning and execution with action costs. As in $(ACE)^2$, we posit $n$ computational processes, all attempting to solve the same problem. To model the execution of actions in the real world, we assume that each process $i$ has already computed a prefix of its complete plan, denoted $H_i$ ($H$ for head). That is, $H_i$ is a (possibly empty) sequence of actions from a set of base-level actions $B$. Each action $b \in B$ has a known duration $dur(b)$ and may have a known deadline $D(b)$. The rest of process $i$'s plan, denoted $\beta_i$, is still unknown.

Base-level actions can be executed even before having a complete plan. When a base-level action is executed, any process $i$ for which the executed plan so far is not a prefix of $H_i$, is invalidated. We make the simplifying assumptions that actions are irreversible, uninterruptible, and cannot be executed concurrently with other actions in $B$. Computation, however, may be concurrent with action execution.

Process $i$ can compute the rest of its plan, $\beta_i$, when given sufficient computation time. We assume that the base-level actions in $\beta_i$ can be executed only when process $i$ terminates. Also when the execution of a remainder $\beta_i$ begins, any other process $j \neq i$ is invalidated. This assumption corresponds to each process representing a different subtree in a forward state-space search in which different actions modify the state in different ways. Thus, executing one remainder modifies the state in a way that is incompatible with all the other remainders.

Since $\beta_i$ is unknown until process $i$ terminates and delivers the rest of its plan, we assume a known distribution $R_i$ on the duration of the remainder $\beta_i$. That is, if we extend the $dur$ function to have an action sequence $S$ as input, such that $dur(S) = \Sigma_{i=1}^{|S|} dur(S[i])$, then $R_i$ is a distribution on $dur(\beta_i)$.

As in the $(ACE)^2$ model, every process has a (possibly unknown) overall deadline; we assume a known distribution of a random variable $X_i$, over $d_i$ the deadline for process $i$. We say that the execution of a solution delivered by process $i$ is *timely* only if the execution of the remainder $\beta_i$ begins in time to complete before the deadline $d_i$. That is, denoting by $start(\beta_i)$ the time at which the execution of $\beta_i$ starts, the execution of the solution is timely only if $start(\beta_i) \leq d_i - dur(\beta_i)$. Since both $d_i$ and $dur(\beta_i)$ are random variables before process $i$ terminates, $start(\beta_i)$ is also a random variable, which we call the *induced deadline* for process $i$ and denote by $D_i$. By definition, $D_i = X_i - R_i$. Thus, we can assume that for every process $i$, the induced deadline $D_i$ is given, and ignore $X_i$ and $R_i$ henceforth.

In addition, every process $i$ has a performance profile described by a CDF $M_i(t)$, the probability that process $i$ will terminate given an accumulated computation time of $t$ or less. We also have a known distribution $C_i$ over solution costs for process $i$ and a cost of failure $c_f$. The true values of the induced deadline and the plan's cost for process $i$ are revealed only when process $i$ terminates. The cost of a plan of an incomplete process, as well as a completed process that has failed to find a timely solution, is assumed to be $c_f$.

We can now define the CoPEC problem, as follows. We have a set of base-level actions $B$, and a cost of failure $c_f$. Given $n$ processes, each with a sequence $H_i$ of actions from $B$, a performance profile $M_i$, an induced deadline distribution $D_i$, and a plan cost distribution $C_i$, the objective is to find a policy for allocating the computation time among the $n$ processes and executing base-level actions from the set $B$, as well as optionally stopping deliberation and executing a complete plan already computed by one of the processes, so as to minimize the expected cost of the executed plan.

**Example 1.** Extending the example from the introduction, (agent needing to get to the airport in time for a flight), we have two candidate plans (processes): process 1 for the train plan, and process 2 for the taxi plan. Suppose that a unit of time is one minute, and that the agent needs to be in terminal A at the airport 30 minutes from now. Let the cost of failing, i.e. not reaching terminal A in 30 minutes and missing the flight, is \$500 (cost of the flight). The train leaves in 8 minutes, the ride takes 20 minutes, and costs \$55. Suppose

the planner has already found that the first action in the train plan is to walk to the station (which takes 5 minutes), but has not yet found what to do at the end of the ride. It may require an additional 10 minutes to walk to terminal A (say with probability 0.2), or terminal A may be adjacent to the train station (with probability 0.8), which requires no transit time. The taxi can arrive in 5 minutes from the moment it is called, takes 18 minutes to reach terminal A, and costs 30\$. The first action of the taxi plan is calling a taxi, which takes one minute, and the second action is to wait for the taxi for 5 minutes. However, the planner has not yet determined what to do at the end of the ride. At the end of the taxi ride, there is a payment, which takes two minutes. Suppose that the remaining time for finding the rest of the taxi plan is 4 minutes, and the remaining time for finding the rest of the train plan is either 2 or 7 minutes, each with probability 0.5.

Translating this into a CoPEC problem instance, base-level actions $B = \{walk, call, wait, train, taxi\}$, with $dur(walk) = 5$, $dur(call) = 1$, $dur(wait) = 5$, $dur(train) = 20$, and $dur(taxi) = 18$. Failure costs $c_f = 500$. The process plan prefixes are $H_1 = [walk, train]$, $H_2 = [call, wait, taxi]$. The performance profiles are distributed $M_1 \sim [0.5 : 2, 0.5 : 7]$ and $M_2 \sim [1 : 4]$. The deadlines for both processes are known, that is, $X_1 = X_2 = 30$ with probability 1. The distributions over the duration of the remainder of the plans are $R_1 \sim [0.8 : 0, 0.2 : 10]$ for $\beta_1$, and $R_2 \sim [1 : 2]$ for $\beta_2$. Therefore, the induced deadlines are distributed $D_1 \sim [0.8 : 30, 0.2 : 20]$ and $D_2 \sim [1 : 28]$. The distributions over the plan costs are $C_1 \sim [1 : 5]$ and $C_2 \sim [1 : 30]$. An optimal policy here is to first run process 1 for 2 minutes. If process 1 terminates and reveals that its deadline is 30, then start walking to the train station and proceed with the train plan. If process 1 does not terminate, or reveals that its deadline is 20, then start executing the actions in $H_2$ (call a taxi), run process 2 for 4 minutes, and proceed with the taxi plan. The expected cost of this policy is $E = 0.5 \cdot 0.8 \cdot 5 + 0.6 \cdot 30 = 20$.

### 3.2 The CoPEC MDP

Following (Shperberg et al. 2019), we formulate a discrete-time version called DCoPEC, allowing us to model the problem as an MDP, similar to the one defined for $(ACE)^2$. We define $m_i(t) = M_i(t) - M_i(t-1)$, the probability that process $i$ terminates after exactly $t$ steps of computation, and $d_i(t) = D_i(t) - D_i(t-1)$, the probability that the deadline for process $i$ is exactly at time $t$.

The state variables are the wall clock time $T$, and one state variable $T_i$ for each process, with domain $\mathbb{N}$, which represents the cumulative time assigned to process $i$ so far. In addition, we have state variables $ct_i$ and $dl_i$, the cost and deadline (respectively) of each process that has completed its computation. We also have state variables $rt$ and $ne$. The variable $rt$ denotes the remaining time to execute an action, in case some action is currently being executed. The variable $ne$ denotes the number of actions that were already executed, including the currently executed action, if such exists. We also have a special terminal state DONE. As in the $D(ACE)^2$ MDP, the $ct_i$ and $dl_i$ variables are irrelevant for incomplete processes, and the time assigned to a process is

irrelevant to completed processes, thus the state space can be stated as:

$$S = \{\text{DONE}\} \cup \Big( dom(T) \times dom(rt) \times dom(ne)$$

$$\times \mathop{\times}_{1 \leq i \leq n} \big( dom(T_i) \cup (dom(ct_i) \times dom(dl_i)) \big) \Big)$$

The initial state $S_0$ has $T = 0$, $T_i = 0$ for all $1 \leq i \leq n$, $rt = 0$ and $ne = 0$. We use the notation $T[S_0] = 0$ and $T_i[S_0] = 0$ (i.e. state variable as a function of the state) as a shorthand to denote this.

There are three types of actions in the MDP. The first type, denoted $a_i . i \in [1, n]$, assigns the next unit of computation time to an unterminated process $i$ that has not already failed. The second type, denoted $g_i$, gives the plan computed by completed process $i$ the go-ahead to execute, and transitions into a terminal state. Note that for every process $i$, either $a_i$ or $g_i$ is applicable, but not both. The third type is executing the base-level action $b \in B$. This can be thought of as committing to execute action $b$ in the following $dur(b)$ time units. Action $b$ is applicable only if there is no other action executing at the same time.

The transition distribution is determined by the executed base-level actions and by the $M_i$ and $D_i$ distributions. If a process $i$ completes its computation in the transition from state $S$ to state $S'$, then $ct_i[S']$ and $dl_i[S']$ are assigned according to the actual deadline and cost of the solution obtained by process $i$. In this case, the time at which the execution of the solution delivered by process $i$ will end (given that the execution starts now) is:

$$l_i(S, S') = T[S'] + rt[S'] + dur(H_i[ne[S]...|H_i|])$$

where $H_i[j...k]$ denotes a subsequence of sequence $H_i$ from $j$ to $k$, inclusive. That is, $l_i(S, S')$ equals the time after allocating one time unit to process $i$, plus the remaining time for executing the current base-level action (if such exists), plus the duration of the remaining tail of the prefix $H_i$. Then, when transitioning from state $S$ to $S'$ by applying action $a_i$ (which is applicable only if process $i$ has not terminated):

- The current time $T[S'] = T[S] + 1$.
- $rt[S'] = max\{0, rt[S] - 1\}$.
- The computation time of every other process remains unchanged, that is $\forall j \neq i : T_j[S'] = T_j[S]$.
- The probability that process $i$'s computation completes in this transition (given that it has not previously terminated) is $P_{term} = \frac{m_i(T_i[S]+1)}{1 - M_i(T_i[S])}$. Therefore, with probability $1 - P_{term}$, process $i$ does not complete and we have $T_i[S'] = T_i[S] + 1$.
- Conversely, with probability $P_{term}$, process $i$ completes. In this case, $ct_i[S']$ and $dl_i[S']$ are assigned values according to distributions $C_i$ and $D_i$, respectively. For example, in the independent cost case, for all $x \leq d_{max_i}$, $dl_i[S'] = x$ with probability $d_i(x)$. If $x < l_i$, then $ct_i[S'] = c_f$ (with probability 1), otherwise, for all $y \in Supp(C_i)$, $ct_i[S'] = y$ with probability $C_i(y)$.

The state variables for other processes remain unchanged. Denote by $d_{max_i}$ the last deadline for process $i$, i.e. the smallest $t$ for which $D_i(t) = 1$. The reward for being in state $S$ and executing action $a_i$ such that a solution delivered by process $i$ can be executed entirely before the last deadline for process $i$, that is $T[S] + rt[S] + dur(H_i[ne[S]...|H_i|]) < d_{max_i}$, is always 0. However, when $a_i$ is applied in state $S$ such that $T[S] + rt[S] + dur(H_i[ne[S]...|H_i|]) \geq d_{max_i}$, the reward is $-c_f$. In the latter case, transition into $S' = \text{DONE}$ with probability 1. This exception is in order to avoid useless allocation of time to processes that cannot find a timely plan, as well as infinite allocation sequences.

When applying action $b$ in state $S$ (which is applicable only if $rt[S] = 0$), that is, declaring that action $b \in B$ will be executed in the next $dur(b)$ time units, then if $T[S] + dur(b) > D(b)$, the transition is to terminal state $S' = \text{DONE}$ with probability 1. The reward in this case is $-c_f$. Otherwise, transition to state $S'$ such that:

- The current time does not change, $T[S'] = T[S]$.
- $rt[S'] = dur(b)$ and $ne[S'] = ne[S] + 1$,
- For any process $i$ whose prefix $H_i$ is not compatible with $b$, i.e. $H_i[ne[S]] \neq b$, assign $dl_i[S'] = -1$ and $ct_i[S'] = c_f$. For any other process $i$, for which $H_i[ne[S]] = b$, the accumulated time is preserved, i.e. $T_i[S'] = T_i[S]$.

The reward in this case is 0.

When applying action $g_i$ in state $S$ (applicable only if process $i$ has terminated and the variables $dl_i[S]$, $ct_i[S]$ have been assigned), that is, executing the plan found by process $i$, the transition is always to terminal state $S' = \text{DONE}$. The reward in this case is $-ct_i[S]$ if $dl_i[S] \geq T[S] + rt[S] + dur(H_i[ne[S]...|H_i|])$ and $-c_f$ otherwise.

## 4 Theoretical Examination of CoPEC

Shperberg et al. (2020a) showed that, for the general case, with unrestricted dependencies, optimally solving the $(ACE)^2$ problem is PSPACE-hard. Since CoPEC is a generalization of $(ACE)^2$, unrestricted CoPEC problem is also PSPACE-hard. We thus examine special cases to see whether they have lower complexity. In particular, we consider special cases where the deadlines and costs are known, and the number of different plan costs is restricted. If the optimal policy can be described by a simple sequence of actions (called "linear"), this would simplify solution algorithms. Unfortunately, even under additional restrictions, the optimal policy may be non-linear.

**Theorem 4.1.** *Even for the special case of DCoPEC where all plan costs are 0, except for one plan with cost 1, and a cost of failure greater than 1, the optimal solution requires a conditional (i.e. non-linear) policy.*

*Proof.* Consider processes $\{1, 2, 3\}$ with costs $C_1 = 1$, $C_2 = 0$, $C_3 = 0$ and with deadlines $d_1 = 2$, $d_2 = 2$, $d_3 = 3$. Let the prefixes of the processes be $H_1 = H_2 = [b]$, where $b$ is a base-level action with $dur(b) = 1$ and $deadline(b) = \infty$, and $H_3 = \emptyset$. Let the completion-time distributions be $m_1 = [0.9 : 1, 0.1 : 10]$, $m_2 = [0.1 : 1, 0.9 : 10]$ and $m_3 = [0.2 : 2, 0.8 : 10]$. That is, processes 1 and 2 have

some probability to complete their computation in one time unit, otherwise they fail. Process 3 needs two time units to complete its computation successfully, otherwise it fails. Finally, let the cost of failure be $c_f = 10$.

It can be shown that the only optimal policy $P^*$ is contingent: first run process 1 for one time unit. If process 1 terminates, then execute base-level action $b$ and then run process 2. Since base-level action $b$ was executed before the deadline of processes 1 and 2 passed, the expected cost in this branch is $0.1 \cdot 0 + 0.9 \cdot 1 = 0.9$, because if process 2 succeeds we get a plan with cost 0 (do action $g_2$), and if process 2 fails we get a plan with cost 1 (do action $g_1$). If process 1 fails, then run process 3 for two time units. The expected cost in this branch is $0.2 \cdot 0 + 0.8 \cdot 10 = 8$, because if process 3 succeeds we get a plan with cost 0 (do action $g_3$), and if process 3 fails we fail to find a plan altogether and get the cost of failure 10. Thus, the expected cost of policy $P^*$ is $E[C(P^*)] = 0.9 \cdot 0.9 + 0.1 \cdot 8 = 1.61$. □

### 4.1 Special Case: One Remaining Process

Following Shperberg et al. (2020a), we consider the case where there is only one running process, w.l.o.g. we assume it is process 0, and $n-1$ terminated processes that already delivered plans with known induced deadlines $d_j$ and known costs $c_j$, with $j \in [1, n-1]$. We begin by showing that in this simplified case, there exist optimal policies that have special properties, which simplify the analysis.

We only need to consider policies that start from the initial state $S_0$. Thus, a policy can be represented as an and-tree rooted at state $S_0$. The possible actions can be represented as edges from each state node, that lead to chance nodes with all the next possible states as children (see example 1).

Observe that a state in which all $n$ processes are terminated (and have known induced deadlines $d_j$ and known costs $c_j$) or failed (i.e. have deadline $d_j = -1$ and cost $c_j = c_f$), can be considered a terminal state. The reward in such a state would be minus the minimal plan cost $\min_{1 \leq j \leq n}\{c_j\}$ for which the deadline $d_j$ has not passed. This is equivalent to committing to the best timely plan $i$ (using action $g_i$).

**Definition 4.1.** *A policy tree is called a linear policy if every chance node leads to at most one non-terminal state node with non-zero transition probability.*

Such policy trees are called linear because they can be represented by a sequence $A$ of meta-level and base-level actions, where $A[t] = a_i$ means 'assign time unit $t$ to process $i$' and $A[t] = b$ means 'execute base-level action $b \in B$ starting from time $t$'.

**Theorem 4.2.** *In the case of CoPEC with only one running process, every legal policy is a linear policy.*

*Proof.* Let $\sigma$ be a legal policy, and consider some chance node $N$ in $\sigma$. There are three cases: **(i)** The edge leading to $N$ represents a base-level action $b \in B$. In this case the transition is deterministic: $N$ has a single child. **(ii)** The edge leading to $N$ represents a go-ahead action $g_i$. Here the transition is also deterministic (to terminal state DONE with probability 1). **(iii)** The edge leading to $N$ represents an allocation action $a_i$. Note that all processes $j \in [1, n-1]$

have terminated, so only $a_0$ is allowed. In this case, with probability $1 - P_{term}$, $N$ transitions into a state where the running process 0 does not terminate. Otherwise, $N$ transitions into states $S$ where process 0 terminated (either with cost $c_f$ or with some cost $c \in Supp(C_i)$). Since the remaining $n-1$ processes have also terminated, $S$ are all terminal states. Therefore, $N$ has at most one non-terminal child. □

Given a linear policy $\sigma$, we denote by $BLA(\sigma)$ the sequence of base-level actions executed according to $\sigma$.

**Theorem 4.3.** *In CoPEC with only process $p_0$ unterminated, there exists an optimal policy $\sigma$ s.t. $BLA(\sigma) \subseteq H_0$.*

*Proof.* Let $\sigma$ be an optimal (linear) policy. If $BLA(\sigma) \subseteq H_0$, then we are done. Otherwise, let k be the first index such that $BLA(\sigma)[1...k]$ is not a prefix of $H_0$. Denote $b_k = BLA(\sigma)[k]$, and let $t$ be the time at which $b_k$ is executed according to $\sigma$, that is $\sigma[t] = b_k$. Note that once $b_k$ is executed, the running process 0 is invalidated, and since all other processes are either completed or failed, the policy $\sigma$ transitions into a terminal state with probability 1. Thus, $b_k$ is the last action in policy $\sigma$.

Let $i$ be a valid plan with the lowest cost before $b_k$ is executed. Consider policy $\sigma' = \sigma[1...t-1] \cdot [g_i]$. We now show that policy $\sigma'$ is optimal. Let $j$ be a valid plan with the lowest cost after $b_k$ is executed. Note that $j$ is also a valid plan before $b_k$ is executed, thus, $c_i \leq c_j$. Since the only difference between $\sigma'$ and $\sigma$ is committing to plan $i$ (with cost $c_i$) instead of executing action $b_k$ and then committing to a plan with cost $c_j$, we get that $E(\sigma') = E(\sigma) - c_j + c_i \leq E(\sigma)$. Thus, $\sigma'$ is optimal and $BLA(\sigma') \subseteq H_0$. □

### 4.2 One Process and Known Deadlines

We simplify this case even further and assume that process 0 has a known deadline $d_0$. We show that for this simplified case, there exists an XP algorithm that computes an optimal policy in time $O(n^{|H_0|+2})$, as follows.

In order to find an optimal policy, we need to consider which base-level actions to execute and when to execute each action. In addition, we need to consider how much computation time to give to running process 0. By theorem 4.3, in CoPEC with only one running process (even with unknown deadline), there exists an optimal policy $\sigma$ such that $BLA(\sigma) \subseteq H_0$. Thus, we need to consider only sequence $H_0$, or a sub-sequence of $H_0$, rather than all prefixes $H_j$.

We call a mapping from a sequence $H_i$ to the action execution start times an *initiation function*, and denote it $I_i$. That is, for every $1 \leq k \leq |H_i|$, $I_i(H_i[k])$ is the time at which base level action $H_i[k]$ is executed. Given an initiation function $I_i$, we can compute the exact time, the "effective deadline" $d_j^{eff}(I_i)$, at which each process $j$ will be invalidated (according to the deadline $d_j$ and $I_i$). There are two cases in which process $j$ can be invalidated (before its induced deadline $d_j$):

**(i)** $H_i$ is not a prefix of $H_j$. In this case, process $j$ will be invalidated at the time the first base-level action $b \in H_i \setminus H_j$ is executed.

**(ii)** $H_i$ is a prefix of $H_j$, but there exists an index $m$ such that the execution of base-level action $H_j[m]$ must start before time $I_i(H_i[m])$, otherwise process j will be tardy. In this case, process $j$ will be invalidated at the time the execution of action $H_j[m]$ must start, that is, at time $d_j - dur(H_j[m...|H_j|])$.

Therefore, denoting by $m$ the first index for which

$$(H_i[m] \neq H_j[m]) \vee (I_i(H_i[m]) > d_j - dur(H_j[m...|H_j|])) \quad (3)$$

the effective deadline for process $j$ is:

$$d_j^{eff}(I_i) = min\left( I_i(H_i[m]), d_j - dur(H_j[m...|H_j|]) \right) \quad (4)$$

The effective deadline of running process 0 is simply $d_0$.

To compute the optimal policy we may have to examine all possible initiation functions (mappings from the prefix $H_0$ to possible action execution start times) $I_0$. However, we show that all but $O(n^{|H_0|})$ of them are dominated

Given a process $i$ with a known deadline $d_i$, and some base-level action $b \in H_i$, we denote by $LST_i(b)$ the latest start time of $b$ according to deadline $d_i$. That is, let $k$ be the index such that $H_i[k] = b$, then $LST_i(b) = d_i - dur(H_i[k...|H_i|])$ (see example 2). We get:

**Theorem 4.4.** *In the case of CoPEC where there is only one running process with a known deadline, there exists an optimal policy $\sigma$ such that $BLA(\sigma) \subseteq H_0$ and for every base-level action $b \in BLA(\sigma)$ there exists a process $i$ such that $b$ is executed at time $LST_i(b)$.*

*Proof.* From theorem 4.3, there exists an optimal (linear) policy $\sigma$ such that $BLA(\sigma) \subseteq H_0$. Let $k$ be the maximal index in the sequence $BLA(\sigma)$ such that base-level action $b_k = BLA(\sigma)[k]$ is executed at time $t_k \neq LST_i(b_k)$ for all $i$. Let $P_k$ be the set of processes that are valid after base-level action $b_k$ is executed, and let $j \in P_k$ be a process with a minimal deadline $d_j$. Since $b_k$ is executed at time $t_k \neq LST_j(b_k)$ and since $j$ is still valid after the execution of $b_k$, we deduce that $t_k < LST_j(b_k)$. Let $\sigma'$ be the policy where we swap base-level action $b_k$ with the actions that follow $b_k$ until $b_k$ is executed at time $LST_j(b_k)$ (if there are not enough actions that follow $b_k$, we just add computation actions $a_0$). That is, $\sigma' = \sigma[1...t_k - 1] \cdot \sigma[t_k + 1...LST_j(b_k) - 1] \cdot [b_k] \cdot \sigma[LST_j(b_k) + 1...|\sigma|]$.

We now show that $\sigma'$ is a legal policy (i.e. there are no overlaps between base-level actions). If $b_k$ is the last base-level action executed according to $\sigma$, since $b_k$ is executed at time $LST_j(b_k) > t_k$ according to $\sigma'$, there are no overlaps between $b_k$ and any other base-level action, and therefore, $\sigma'$ is legal. Otherwise, let $b_{k+1} = BLA(\sigma)[k + 1]$ be the base-level action that is executed after $b_k$ according to $\sigma$. Note that, since the index $k$ is maximal, there exists a process $i \in P_k$ such that $b_{k+1}$ is executed at time $t_{k+1} = LST_i(b_{k+1})$. As $d_j$ is a minimal deadline among the deadlines of all processes $i \in P_k$, we can deduce that $t_{k+1} = LST_i(b_{k+1}) \geq LST_j(b_k) + dur(b_k)$. Therefore, the subsequence $\sigma[t_k + 1...LST_j(b_k) + dur(b_k)]$ consists only of computation actions $a_0$, and thus $BLA(\sigma') = BLA(\sigma)$. In addition, since $b_k$ is executed at time $LST_j(b_k) > t_k$ according to $\sigma'$, and since there are no other base-level actions

executed at time $LST_j(b_k) \leq t \leq LST_j(b_k) + dur(b_k)$, there are no overlaps between $b_k$ and any other base-level actions executed according to $\sigma'$. Thus, $\sigma'$ is a legal policy.

Finally, since according to $\sigma'$, $b_k$ is executed at time $t'_k = LST_j(b_k) \leq LST_i(b_k)$ for all $i \in P_k$, then all the processes that are valid after the execution of $b_k$ according to $\sigma$, are also valid after the execution of $b_k$ according to $\sigma'$. Therefore, since the rest of policy $\sigma$ remains unchanged, we get that the expected cost $E(\sigma') = E(\sigma)$, hence, $\sigma'$ is an optimal policy.

Repeating these steps for all base-level actions $b \in BLA(\sigma)$, we get an optimal policy as required. $\square$

By theorem 4.4, it suffices to compute all the legal mappings from $H_0$ by considering the latest start time of the base-level actions according to the deadlines $d_i$ for all $i \in [0, n-1]$ (see example 2). Note that there are at most $n^{|H_0|}$ such mappings.

**Example 2.** Consider processes $\{0, 1\}$ with prefixes $H_0 = H_1 = [b_1, b_2]$, where $dur(b_1) = dur(b_2) = 1$, and with deadlines $d_0 = 4$, $d_1 = 2$. Then $lst_0(b_2) = 3$, $lst_0(b_1) = 2$, $lst_1(b_2) = 1$ and $lst_1(b_1) = 0$. Thus, we return the following mappings: $\{(b_1, 2), (b_2, 3)\}$, $\{(b_1, 0), (b_2, 1)\}$, $\{(b_1, 0), (b_2, 3)\}$. Note that the mapping $\{(b_1, 2), (b_2, 1)\}$ is illegal since action $b_1$ needs to be executed after the subsequent action $b_2$.

Once we compute all the initiation functions, we can compute the effective deadlines $d_j^{eff}(I_0)$ for all processes $j$, for every initiation function $I_0$. Then, given effective deadlines $d_j^{eff}(I_0)$, we can simply use them (instead of the induced deadlines $d_j$) in the equation proposed by Shperberg et al. (2020a). Factoring the fact that the deadlines are known, we never schedule computation after the respective plan is bound to be tardy, so we get:

$$E_{\pi_{o,k}}(d^{eff}(I_0), 0) = (1 - M_0(d_k^{eff}(I_0))) \cdot c_k +$$

$$\sum_{j=2}^{k} (s_0(d_j^{eff}(I_0), 0) - s_0(d_{j-1}^{eff}(I_0), 0)) E[min(C_0, c_j)] \quad (5)$$

where $s_0(t, t_d)$ is the probability that process 0 will find a timely plan given $t$ time units starting at time $t_d$.

In the algorithm, for each such possible initiation function $I_0$, we compute the effective deadlines $d_j^{eff}(I_0)$ using equation 4. Given the effective deadlines, we iterate over $k$ to find the expected cost of a best policy (i.e. a policy with minimal expected cost) using equation 5. Finally, we return the policy with the minimal expected cost that was found.

## 5 Algorithms for the General Case

We propose several algorithmic schemes for CoPEC. In all schemes below that require a known deadline, we use as the deadline $d_i$ the minimal value in the support of $D_i$, though other values can be chosen such as the expectation.

### 5.1 $Demand\text{-}Execution_A$

The $Demand\text{-}Execution_A$ scheme uses any $(ACE)^2$ algorithm $A$, and adapts it to CoPEC. Given a CoPEC problem instance, at each decision point we use algorithm $A$ to

choose which process $i$ is given computation time. Then, we check whether the next base-level action in the prefix $H_i$ must be executed now in order for process $i$ to be non-tardy. That is, let $ind$ be the index of the first base-level action in prefix $H_i$ that was not yet executed, and let $T$ be the wall clock time. Given a deadline $d_i$, the latest start time of action $H_i[ind]$ is $lst_i(H_i[ind]) = d_i - dur(H_i[ind...|H_i|])$. Thus, if $lst_i(H_i[ind]) \leq T$, return base-level action $H_i[ind]$, otherwise return computation action $a_i$.

## 5.2 $Min\text{-}LET_{DAG}$

The $Min\text{-}LET_{DAG}$ algorithm is based on the algorithm for the special case where there is one running process and $n-1$ completed processes. We compute an initiation function $I_i$ for each process $i$ by setting each base level action $b$ in $H_i$ to be at the latest execution time at which it must be executed according to the deadline $d_i$ and to the deadline of $b$, $D(b)$. Pseudo-code for computing the initiation function $I_i$ for process $i$ is given in Algorithm 1.

---

**Algorithm 1:** Compute the latest execution time of base-level actions

   **Input:** $H_i$ - the head of process $i$, $d$ - a value from the support of $D_i$
   **Output:** An initiation function $I_i$
**1 Function** $compute\_init\_func$ ($H_i$, $d$) **:**
**2**     Let $I = \{\}, t = d$
**3**     **for** $j = length(H_i) \ldots 1$ **do**
**4**        $t = \min(t - dur(H_i[j]), deadline(H_i[j]) - dur(H_i[j]))$
**5**        $I[H_i[j]] = t$
**6**     **return** $I$

---

As before, given an initiation function $I_i$, we can compute the effective deadlines $d_j^{eff}(I_0)$ (equation 4) for all processes $j$, and reduce the instance into an $(ACE)^2$ instance. Then, for each process $i$, we compute the effective deadlines $d_j^{eff}(I_i)$ of all the processes $j$ according to the initiation function $I_i$. The $Min\text{-}LET_{DAG}$ algorithm iteratively allocates time to the process $i$ that maximizes $Q_i(d^{eff}(I_i), t_d)$, and executes base-level actions according to the initiation function $I_i$. See Algorithm 2 (one iteration).

## 5.3 MPP

We also implemented a Most Promising Process (MPP) algorithm, where in each iteration we schedule the process with the lowest expected cost. The expected cost of a process $i$ is computed according to the $M_i$, $D_i$ and $C_i$ distributions, as follows. For every value $t \in Supp(M_i)$, compute the probability $p_{term}$ that process $i$ will terminate in $t$ time units. Then for each deadline $x \in Supp(D_i)$, if process $i$ will miss the deadline $x$ after running for $t$ time units, iteratively add $p_{term} \cdot d_i(x) \cdot c_f$ to the expected cost. Otherwise, if process $i$ will meet deadline $x$, iteratively add $p_{term} \cdot d_i(x) \cdot C_i(y) \cdot y$ for every cost $y \in Supp(C_i)$. The pseudo-code for computing (recursively) the expected cost of the plan delivered by a process $i$ is appears as Algorithm 3.

---

**Algorithm 2:** $Min\text{-}LET_{DAG}$ Outline

   **Input:** $S$ - a state of an CoPEC problem instance
   **Output:** A base-level action or a meta-level action
**1** Let $I = (), Q = ()$
**2** **for** $j = 1, ..., n$ **do**
**3**     $d = \min(Supp(D_j))$
**4**     $append(I, compute\_init\_func(H_j, d))$
**5**     $d^{eff} = compute\_effective\_deadlines(I_j)$
**6**     $append(Q, Q_i(d^{eff}, t_d))$
**7** $i = argmax_{j \in [1,n]} Q_j; \quad ind = ne[S]$
**8** **if** $I_i(H_i[ind]) \leq T[S]$ **then**
**9**     **return** $H_i[ind]$
**10 else**
**11**     **return** $a_i$

---

**Algorithm 3:** Expected plan cost for process $i$

   **Input:** $S$: state of an CoPEC instance, $i$: a process#
   **Output:** The expected cost of plan $i$
**1 Function** $expected\_cost$ ($S$, $i$) **:**
**2**     **if** $is\_empty(M_i)$ **then**
**3**        **return** $c_f$
**4**     $t = draw(M_i)$     ▷ draw the minimal value from $M_i$ $p_{term} = \frac{m_i(t)}{1 - M_i(t-1)}$;
**5**     $cost = 0$;
**6**     **for** $x \in Supp(D_i)$ **do**
**7**        **if** $T[S] + t - T_i[S] > x$ **then**
**8**           $cost = cost + p_{term} \cdot d_i(x) \cdot c_f$;
**9**        **else**
**10**           **for** $y \in Supp(C_i)$ **do**
**11**              $cost = cost + p_{term} \cdot d_i(x) \cdot C_i(y) \cdot y$;
**12**     $M_i = remove(M_i, t)$     ▷ remove $t$ in $Supp(M_i)$ $cost = cost + (1 - p_{term}) \cdot expected\_cost$ ($S$, $i$) ;
**13**     **return** $cost$

---

## 6 Empirical Evaluation

Data for our experiments uses weighted 15-puzzle instances: action cost for a tile is the number on the tile. To create CoPEC problem instances, we first solved 60,000 puzzle instances using A*, with $h$ =weighted Manhattan distance. For each puzzle instance, we recorded the h-value of the initial state, number of nodes expansions required by A* from the initial state, and length and cost of the path found by A*. Then, we created three PMF and CDF histograms for each initial h-value: the required number of expansions, the optimal solution length, and the optimal solution cost.

Given the three histograms, we create a CoPEC problem instance of $n \in \{2, 5, 10, 20\}$ processes as follows. We drew a random puzzle instance, and ran A* until the open list contained at least $n$ search nodes. We chose the first $n$ nodes in the open list, recording their h-value. Each chosen node $i$ becomes an CoPEC process, where $M_i$ is the node expansion CDF histogram corresponding to node $i$ h-value $h(i)$, $C_i$ is

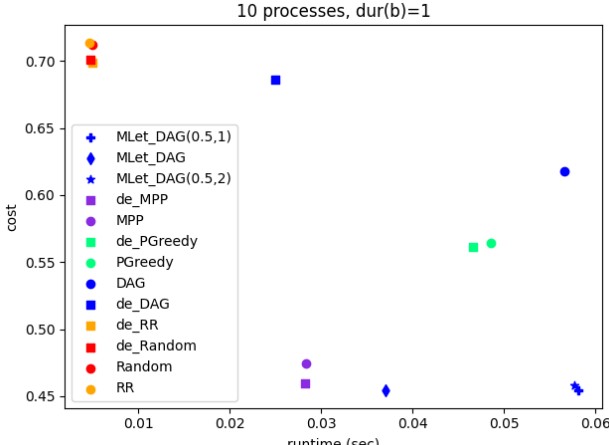

Figure 1: Avg. norm. cost vs. runtime: n=10, $dur(b) = 1$

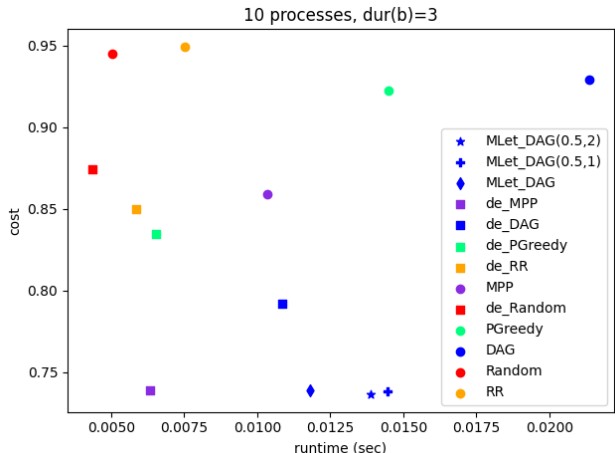

Figure 2: Avg. norm. cost vs. runtime: n=10, $dur(b) = 3$

the solution cost PMF histogram corresponding to $h(i)$, $R_i$ (the distribution over the duration of the remainder of plan $i$) is the solution length CDF histogram corresponding to $h(i)$ multiplied by the actions duration, and the prefix $H_i$ is the sequence of actions leading to node $i$ from the initial state. To get a deadline distribution $X_i$, we also recorded the non-weighted Manhattan distance heuristic value for each search node $j$, we denote it $h'(j)$. Then, we use the (known) deadline distribution $X_i = 4 \cdot h'(i)$ for each chosen node $i$. Recall that the induced deadline distribution is defined as $D_i = X_i - R_i$, thus, even though $X_i$ is known, $D_i$ is unknown as $R_i$ is unknown. For the cost of failure $c_f$, we used the maximal cost among all $C_i$ distributions, multiplied by 2. Finally, we assumed that each base-level action (i.e. move up, down, left or right) requires $l \in \{1, 2, 3\}$ time units to complete. For each such duration $l$ we created a separate CoPEC instance. The empirical evaluation included 50 CoPEC instances in each such setting.

We implemented the following scheduling algorithms as baselines for the evaluation: Basic Greedy Scheme (P-Greedy, (Shperberg et al. 2019)) from the $(AE)^2$ model; Round Robin (RR) - allocate computation time to processes cyclically; Random - allocate computation time to a random (valid) process at each iteration. We also implemented the MPP algorithm, and the DAG algorithm with $\gamma \in \{0, 0.5, 1, 2, 10\}$ and a delay $t_d \in \{0, 1, 2, 5, 20\}$. Results are reported only for the best-performing parameters.

Our results figures present the average expected cost of an optimal policy found by the algorithms, divided by the cost of failure $c_f$, and the algorithm runtime. The legends are sorted by algorithm performance from lowest (i.e. best) to highest expected cost.

The results suggest that most of the algorithms, even those not executing base-level actions until a complete plan is found, perform well on average when the base-level action duration is 1, because time-pressure is not critical here. However, the performance of "raw" algorithms that do not support concurrent execution, gets worse as time pressure increases. The CoPEC schemes Demand-Execution and Min-Let, on the other hand, perform well even under severe

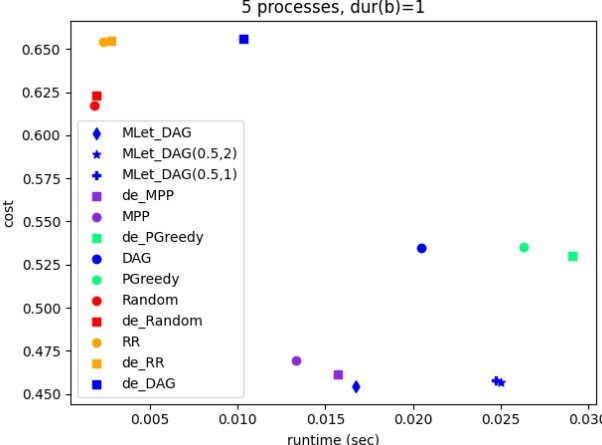

Figure 3: Avg. normalized cost vs. runtime: n=5, $dur(b) = 1$

time pressure, as they execute base-level actions while planning, thus potentially delivering cheaper plans.

The Demand-Execution schemes perform better than the raw algorithms when the time pressure is higher, but worse than the Min-Let schemes. The reason is that the Demand-Execution schemes do not consider the effective deadlines of the processes (i.e. do not consider that processes might be invalidated by the execution of base-level actions). Therefore, when the base-level action duration is 1, the Demand-Execution schemes perform even worse than the raw algorithms, as they execute actions and invalidate good processes (see figures 1 and 3). Whereas the raw algorithms do not invalidate processes, and have enough time to find plan with relatively low expected cost.

The results suggest that $Demand\text{-}Execution_{MPP}$ and $Min\text{-}Let_{DAG}$ have the best performance. When time pressure is low, the $Min\text{-}Let_{DAG}$ versions find policies with lower expected cost, and are almost as fast as $Demand\text{-}Execution_{MPP}$ (see figures 1 and 3). When the time pressure is high, the $Min\text{-}Let_{DAG}$ variants find policies with slightly lower expected cost, but

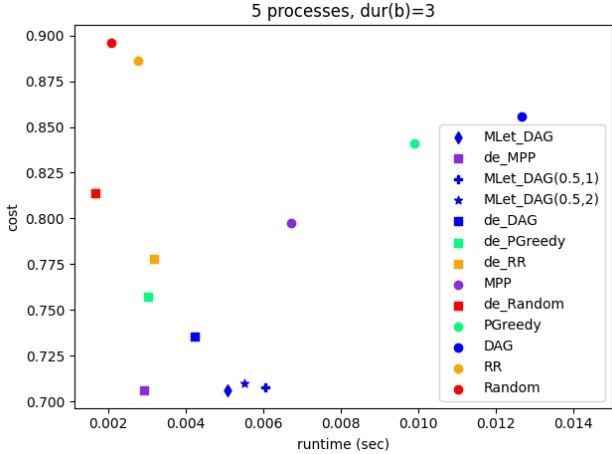

Figure 4: Avg. normalized cost vs. runtime: n=5, $dur(b) = 3$

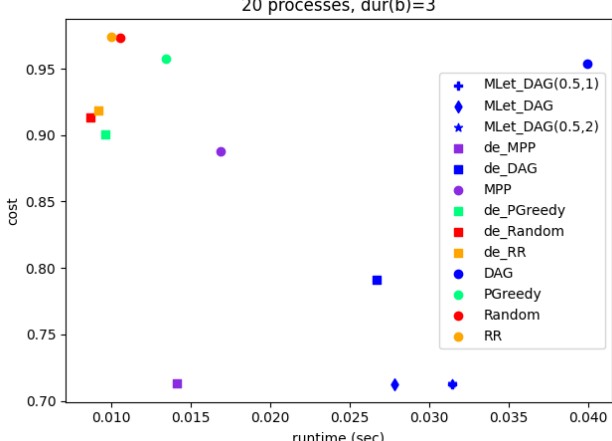

Figure 5: Avg. norm. cost vs. runtime: n=20, $dur(b) = 3$

$Demand\text{-}Execution_{MPP}$ is much faster (see figures 2, 5).

## 7 Conclusion

We defined a formal abstract model for concurrent planning and execution that shows how to trade off the risk of executing incorrect actions against the opportunity of finding cheaper plans. Even the abstract problem is hard, so we examined special cases for which we were able to design greedy algorithms. These seemed to have good performance on problem instances generated from weighted 15 puzzles. This work provides a formal foundation for addressing concurrent planning and execution. However, much work still remains to be done in adapting these algorithms to work inside a situated planner.

## Acknowledgements

This research was supported by grant 2019730 from the United States-Israel Binational Science Foundation (BSF) and grant 2008594 from the United States National Science Foundation (NSF).

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
