# OpenReview forum: "A Formal Model of Concurrent Planning and Execution with Action Costs"
_icaps-conference.org/ICAPS/2022/Workshop/HSDIP — HSDIP 2022_

### Official Review · Reviewer_Jnie · 2022-04-24
**Situated Temporal Planning with a Success/Cost Trade-off**

**Confidence:** 2
**Overall Score:** Accept

**Review:**

The authors present a new approach to solve situated temporal planning tasks (planning tasks where the time to plan may exceed the deadlines of some actions). Previous work just maximized the chance of reaching the goal. Instead a new model is proposed which incorporates a trade off between the chance of succeeding and the cost of the executed actions and many theorems for special cases are proven.

I am certainly lacking background for an in depth review. I regularly had difficulties following you. Nevertheless, the contribution appears interesting and might kick off some interesting discussions. The plenty of examples were helpful.
- Why are all process invalidated when the remainder $\beta_i$ is executed? Because you execute the complete remainder?
- Section 4: "Unfortunately, that is not the case". You mean that the optimal policy is not linear, do you? (It could also refer to "this would simplify solution algorithms").
- Section 6: I am wondering how diverse the set of processes is that you start with. E.g. for n = 2, the difference is only whether tile A or tile B was moved onto the initial blank position. Similar small will be the changes for the other `n`. If the starting state of the processes are quite similar, is it likely that they produce similar plans?
- Figure 1-5: Those figures contain many algorithms which I have difficulties to match to the presented ones. It might be helpful to explain in the experimental section shortly which ones you ran and with which parameters.

Other suggestions:
- if you want to provide an example citation, I suggest to define the following command:
  egcite: \newcommand{\egcite}[1]{\citep[\eg{},][]{#1}}
- if you want to provide multiple citations at once, you can just chain them together in one command:
  \cite{bibkey1, bibkey2}, instead of using two commands within an additional parenthesis.
- References
  - Cashmore et al 2018b:
    - long form of conference (abbreviation used almost everywhere else)
    - pages missing
  - Ruml 2020a:
    - long form + abbreviation of conference used)

---

> ### Author Response · Authors · 2022-04-27
> **Response to the Review**
>
> Many thanks for your helpful suggestions, which we will take into account in the final version if the paper is accepted for the workshop.
>
> Q1: Why are all processes invalidated when the remainder is executed?
>
> A: If we assume that each process represents a different subtree in a forward state-space search in which different actions modify the state in different ways, then executing one remainder modifies the state in a way that is incompatible with all the other remainders.  If one takes a more general view, it could be seen as a simplifying assumption that allows us to avoid modeling the probability that each other process is invalidated.
>
> Q2: You mean that the optimal policy is not linear, do you?
>
> A: Indeed, we meant "unfortunately not linear"
>
> Q3: I am wondering how diverse the set of processes is that you start with [...] is it likely that they produce similar plans?
>
> A: We did not explicitly encourage diversity among the processes.  Our procedure is such that each set of processes will roughly represent the entire search tree as it grows during search.  Had we selected random nodes from a large frontier, there would be a danger that a small number of processes will be obviously superior and hence the metareasoning problem will be trivial.  On the other hand, if many processes share the same prefix, metareasoning is also easy as we can commit to the shared prefix.  Our procedure appears to have yielded problems that help differentiate between the algorithms.

---

### Official Review · Reviewer_8AfB · 2022-04-25
**Concurrent planning and execution with action costs**

**Confidence:** 2
**Overall Score:** Accept

**Review:**

The paper focuses on a problem of planning under time pressure where the agent is executing plan while the planner tries to find the best (continuation of that) plan. In this case, the problem with action costs is considered.

Since I'm not expert in this area, I cannot fully assess the technical contribution, but the paper seems to be technical sound (to the degree I'm able to assess it). On a high-level, I didn't have problem understanding the concept and the achieved results. Topicaly, the paper fits HSDIP.